# Role of Patient-Reported Outcomes in Clinical Trials in Metastatic Colorectal Cancer: A Scoping Review

**DOI:** 10.3390/cancers15041135

**Published:** 2023-02-10

**Authors:** Jan Gerard Maring, Job F. H. Eijsink, Friso D. Tichelaar, Pawida Veluwenkamp-Worawutputtapong, Maarten J. Postma, Daan J. Touw, Jan Willem B. de Groot

**Affiliations:** 1Department of Clinical Pharmacy, Isala, NL 8025 AB Zwolle, The Netherlands; 2Department of Clinical Pharmacy and Pharmacology, University Medical Center Groningen, University of Groningen, NL 9713 GZ Groningen, The Netherlands; 3Department of Health Sciences, University Medical Centre Groningen, University of Groningen, NL 9713 GZ Groningen, The Netherlands; 4Department of Economics, Econometrics & Finance, Faculty of Economics & Business, University of Groningen, NL 9713 GZ Groningen, The Netherlands; 5Department of Pharmaceutical Analysis, Groningen Research Institute of Pharmacy, University of Groningen, NL 9713 GZ Groningen, The Netherlands; 6Isala Oncology Center, Isala, NL 8025 AB Zwolle, The Netherlands

**Keywords:** PROM, colorectal cancer, mCRC, review

## Abstract

**Simple Summary:**

Patient-Reported Outcome Measures (PROMs) provide reports from patients on their own health, quality of life, or functional status associated with their disease, and the care they have received. In treating metastatic colorectal cancer (CRC), it is important to obtain information about the impact of a treatment on various aspects of patients’ lives besides overall survival. We performed a review on the use of PROMs in phase III clinical trials started between 2010 and 2021, evaluating systemic therapy in patients with metastatic CRC. We demonstrate that the quality of reporting on PROMs has increased over the last decade, but is still not optimal. Moreover, PROM results are underreported in studies on metastatic CRC, impeding the optimal incorporation of trial and PROM results into daily clinical practice.

**Abstract:**

Purpose: To perform a scoping review on the use of Patient-Reported Outcome Measures (PROMs) in randomized trials on systemic therapy in patients with metastatic colorectal cancer (mCRC) between 2010 and 2021. Methods: First, a search on clinicaltrials.gov was performed, looking for randomized trials in mCRC. The use of PROMs was analyzed quantitatively. Subsequently, we assessed the completeness of PROM reporting based on the CONSORT PRO extension in publications related to the selected trials acquired using Embase and PubMed. Results: A total of 46/176 trials were registered on clinicaltrials.gov used PROMs. All these trials used validated PROM instruments. The EORTC QLQ-C30 was most frequently used (37 times), followed by the EQ-5D (21 times) and the EORTC QLQ-CR29 (six times). A total of 56/176 registered trials were published. In 35% (*n* = 20), the results of the PROMs were available. Overall, 7/20 (35%) trials documented all items of the CONSORT PRO extension and quality of reporting according to the CONSORT PRO extension was higher than in the period 2004–2012. In 3/20 (15%) of the published trials, the results of PROMs were not discussed nor included in the positioning of the new treatment compared to the reference treatment. Conclusion: When PROMs are used, the quality of reporting on patient-reported outcomes is improving, but this must continue in order to optimize the translation of trial results to individual patient values.

## 1. Introduction

Colorectal cancer (CRC) is the third leading cause of cancer-related death worldwide. In 2020, approximately 1.93 million new CRC cases were diagnosed, and 0.94 million CRC-related deaths occurred worldwide, representing 10% of the global cancer incidence and 9.4% of all cancer caused deaths [1]. The incidence of CRC has been gradually increasing, particularly in developing countries that are implementing the ‘Western’ lifestyle and have adopted screening programs. More men than women are diagnosed with and die from CRC [1].

Survival rates for CRC vary by the stage of disease at diagnosis. The 5-year survival rate of patients with localized stage I-II CRC is 90%, decreases to 72% when lymph nodes are involved (stage III) and drops to 14% in stage IV disseminated disease [2]. From the 1990s, there has been an improvement in the 5-year relative survival rates for CRC in both genders that can be explained by an early diagnosis at initial stages, innovation in the treatment of stage II, III and IV disease with the development of new systemic therapies [3], and a reduction in postoperative mortality [4]. 

Over the past few years, the use of patient-reported outcomes has gained a lot of interest, especially in cancer treatment [5,6]. They represent a patient-centered assessment of health, quality of life and functioning in practice. Patient-Reported Outcome Measures (PROMs) are questionnaires which express a patient’s view on their physical, mental and social functioning, symptoms of disease or health-related quality of life (HRQOL) [7]. By using standardized instruments or questionnaires at various time points, it is possible to collect data on the health condition and well-being of the patient without bias from treating physicians or researchers. These datapoints are different from the clinical outcomes that generally focus on overall or progression-free survival. 

The assessment of treatment efficacy requires a careful and multidimensional approach based on PROMs and functional outcome measures. If trials are to influence clinical care, they must consistently report both clinical and patient-reported outcomes. This is particularly important when treatments may have different impacts on clinical and patient-reported outcomes, for worsening global QoL and symptoms but improved long-term survival. As the treatment of CRC is becoming increasingly complex and more patient-centered, the adoption of PROMs in clinical trials is not only logical but also necessary to obtain information about the impact of a certain treatment on other aspects, including quality of life (QoL) [8], rather than mere overall and progression-free survival. 

Evidence from clinical trials describing treatments in metastatic CRC that include both clinical and patient-reported outcomes combined with an assessment of the quality of reporting of PROM results is limited. A systematic review of randomized controlled trials in CRC patient-reported outcome studies published between 2004 and 2012, found that many of these studies still fail to robustly inform clinical practice since quality of patient-reported outcome reporting is poor [9,10]. We want to explore how PROMs are used in the specific niche of trials on systemic treatment in metastatic CRC and examine how patient-reported outcome research is conducted. The aim of this study is to conduct a scoping review on the use of PROMs (and the type of PROMs) over time in the last decade in clinical trials, evaluating new systemic treatment options in patients with metastatic CRC. Furthermore, we wanted to investigate the relationship between patient-reported and clinical outcomes and, finally, the quality of reporting on patient-reported outcomes and the role of PROMs in the assessment of the value of new treatment modalities and strategies.

## 2. Methods

To obtain an overview of all randomized clinical trials performed in mCRC between 1 January 2010 to 1 January 2022, and in order to generate a detailed picture of PROM instrument selection, gain a contemporary perspective on their planned use in registered studies and to minimize publication bias, we started by performing a search on www.clinicaltrials.gov, with the search terms ‘metastatic colorectal cancer’, ‘colorectal cancer’, ‘cancer’, ‘colorectal’, ‘metastatic‘ and their synonyms, with a search period between 1 January 2010 and 1 January 2022, for recruiting, ongoing and completed phase III randomized trials in adult patients. We included trials on the systemic treatment of stage IV mCRC. We excluded trials that were not randomized; trials that focused on radiation therapy or surgery; trials that focused on pharmacokinetic and pharmacodynamic parameters; trials not published in English; trials on (neo)adjuvant therapy; and trials not investigating any form of systemic therapy (chemotherapy, immunotherapy or targeted therapy).

After the primary search on www.clinicaltrials.gov, we continued to retrieve the peer reviewed publications of these trials. These publications were, after excluding duplicate trial registrations, acquired with the use of the title of the trial and the names of the investigators in Embase and PubMed. The scoping review followed the recommendations of the Preferred Reporting Items for Systematic Reviews and Meta-Analyses (PRISMA) extension for scoping reviews [11]. The protocol has not been registered.

Three reviewers (P.V.W., F.T. and J.W.B.d.G.) independently performed the selection of actual peer-reviewed publications retrieved in two stages: (1) title and abstract screening of each citation; and (2) reviewing the full text of the retained articles. During each stage, disagreements were resolved by consensus or escalated to the research team (J.W.B.d.G., J.F.H.E., J.G.M., D.J.T. and M.J.P.). One reviewer (F.T.) extracted the data into a Microsoft Excel file; the other reviewers (P.V.W., J.W.B.d.G.) verified the extraction file. J.W.B.d.G., J.F.H.E., J.G.M. and F.T. performed the final analyses.

In the primary search on clinicaltrials.gov, we collected data regarding study year, study population and study design. We also collected information describing each outcome assessed in the trial, including details of which outcome constituted the primary endpoint of the trial. Furthermore, we gathered information on study treatment (interventions or treatments employed in the control and experimental arm of the trials) and type of PROMs used. We used the final publications (if available) to identify the number of enrolled patients and number of patients who completed the PROM questionnaires. Additionally, the completeness of reporting was identified according to recommendations from the International Society for Quality of Life Research (ISOQoL) that incorporated these recommendations as a patient-reported outcome extension to the Consolidated Standards of Reporting Trials (CONSORT) statement (the CONSORT PRO extension) [12]. CONSORT PRO consists of 14 items, five patient-reported outcome-specific extensions (used in this manuscript) and nine patient-reported outcome-specific elaborations. The CONSORT PRO extension was calculated for all studies. Each item received a score of 1 or 0, respectively, if rated *yes* or *not*, and an adjusted CONSORT PRO checklist score (i.e., raw score divided by the number of applicable questions) was used to estimate completeness of reporting. In order to compare the patient-reported outcomes and clinical outcomes we used the primary endpoint(s) of each trial to determine whether there was a statistically significant (defined either by a *p*-value or HR depending on the endpoint) improvement as compared to the reference treatment (the clinical outcome was better), a statistically significant deterioration as compared to the reference treatment (the clinical outcome was worse), or neither a statistically significant improvement or deterioration (the clinical outcome was equal). 

The effect of the experimental intervention on patient-reported outcomes was recorded. We classified this as an effect on symptoms alone, an effect on functional scales and global quality of life, or both. Frequently, because of the multidimensional HRQoL questionnaires used, multiple patient-reported outcome domains are analyzed together in longitudinal trials. Over the course of the study, patient-reported outcomes might favor the experimental treatment arm at a given time point and then favor the control treatment arm at a different time point. Therefore, based on the criteria used by Rees et al. [10], the term “broadly” was inserted to account for this possible discrepancy.

Subsequently, we correlated the patient-reported outcomes to the clinical outcomes in the published trials retrieved. We used descriptive statistics to describe the use of PROMs and the correlation of patient-reported outcomes with clinical outcomes in the studies and publications that were identified by our search strategy.

## 3. Results

### 3.1. Data Search

An overview of the search on clinicaltrials.gov and the subsequent search in Pubmed/Embase is shown in Figure 1. 

The search on clinicaltrials.gov led to the identification of 236 clinical trials registered. We excluded six duplicates. We also excluded 54 registered trials because they did not meet the prespecified inclusion criteria (Figure 1). A total of 176 registered trials met the selection criteria. A total of 46 of these trials measured patient-reported outcomes, of these 20 were completed and published (comprising 12,146 patients) [13,14,15,16,17,18,19,20,21,22,23,24,25,26,27,28,29,30,31,32]. In 6 of these 20 publications [15,16,18,23,24,25], the PROMs were reported together with the primary endpoints of the study. 

Table 1 provides an overview of the publications of randomized phase 3 trials on the systemic treatment for metastatic CRC in which PROM results were published.

Of the 20 publications reporting PROMs, 11 involved first-line treatment of mCRC [15,17,18,20,21,22,25,26,28,29,31], including 6852 patients. One publication, including 530 patients, involved second-line treatment [13]. Furthermore, six trials with a total of 4009 patients involved third-line treatment and beyond [14,16,19,23,27,30]. In one publication, patients were treated using second- (*n* = 437 patients) and third-line (*n* = 228 patients) treatments [32]. A total of 967 patients were, therefore, treated using second-line and 4237 patients were treated using third-line treatments. In 3/20 publications (15%), including 1795 patients, the number of patients completing PROMs and the extent of missing patient-reported outcome data was not mentioned [16,24,30]. These publications involved patients treated with third-line treatments and beyond. In the remaining 17 publications, including 10,261 patients [13,14,15,17,18,19,20,21,22,23,25,26,27,28,29,31], PROMs were evaluable in 90% (median, range 60–99%) of patients. Of these patients, 6852 were treated using first-line treatment, 967 with second-line and 2442 with third-line or beyond. 

### 3.2. Quality of Reporting on Patient-Reported Outcomes

In 14 studies [13,14,17,19,20,21,22,26,27,28,29,30,31,32], patient-reported outcomes were reported in a separate manuscript. The mean CONSORT PRO extension checklist score of the studies reported as a separate patient-reported outcome manuscript was 0.79. The mean CONSORT PRO extension checklist score of the studies reporting patient-reported outcomes simultaneously to the primary endpoints was 0.72.

Table 2 depicts the extent to which the included studies adequately reported patient-reported outcomes. The two most frequently reported CONSORT PRO items were the use of well-validated patient-reported outcome instruments (*n* = 20, 100%) and methods for patient-reported outcome data collection (*n* = 20, 100%). The only item that was documented in less than 50% of the trials was reporting a patient-reported outcome hypothesis (*n* = 6, 30%). Overall, 7/20 (35%) trials documented all items of the CONSORT PRO extension. 

Compared to the period between 2004 and 2012, the quality of reporting on patient-reported outcomes in metastatic CRC trials published between 2010 and 2021 has increased with respect to P6a (Evidence of patient-reported outcome instrument validity and reliability should be provided or cited if available) and P20/21 (Patient-reported outcome-specific limitations and implications for generalizability and clinical practice should be discussed). Furthermore, 35% of trials performed between 2010 and 2021 documented all items of the CONSORT PRO extension, whereas 13/66 studies (20%) were considered to have high-quality reporting of the key methodological features of patient-reported outcome design between 2004 and 2012 [10].

### 3.3. The Use of PROMs over Time in Clinical Trials

A total of 46 trials registered on clinicaltrials.gov in the period 2010–2021 intended to assess PROMs and we could retrieve the type of PROMs used in all of them. Only 20 of these trials resulted in publications of the patient-reported outcomes [13,14,15,16,17,18,19,20,21,22,23,24,25,26,27,28,29,30,31,32]. In 21 studies registered on clinicaltrials.gov in the period 2010–2021, one PROM was used; in 19 studies, two PROMs were used; and in six studies, more than two PROMs were used.

The European Organization for Research and Treatment of Cancer general health status and quality-of-life questionnaire (EORTC QLQ-C30) was the most often used PROM (37 times, 80%). The EuroQol-five dimension index questionnaire (EQ-5D) and the European Organization for Research and Treatment of Cancer general health status and quality-of-life questionnaire-Colorectal Cancer Module 29 (EORTC QLQ-CR29) were used 21 and 6 times, respectively. In addition, a large range of alternative PROMs was used, including the MD Anderson Symptom Inventory for gastrointestinal cancer (MDASI-GI), Patients’ Global Impression of Change (PGI-C), Patients’ Global Impression of Severity (PGI-S), European Organization for Research and Treatment of Cancer general health status and quality-of-life questionnaire-Liver Metastases Colorectal 21 (EORTC QLQ-LMC21), 36-Item Short Form Survey (SF-36), SKINDEX-16, National Comprehensive Cancer Network/Functional Assessment of Cancer Therapy Colorectal Cancer Symptom Index-19 (NFCSI-19), Functional Assessment of Cancer Therapy-Colorectal (FACT-C), Dermatology Life Quality Index (DLQI), Geriatric Depression Scale (GDS), Patient-Reported Outcomes Measurement Information System (PROMIS) physical function short form 6a questionnaire, Patient-Reported Outcome version of the Common Terminology Criteria for Adverse Events (PRO-CTCAE), Functional Assessment of Chronic Illness Therapy—Fatigue (FACIT-F), or specifically defined QoL forms. 

There was an increase in the use of the EORTC-QLQ-C30 and EORTC-QLQ-CR29 in trials reporting patient-reported outcomes during the period between 2010 and 2022.

Figure 2 depicts the trends of the utilization of the three most frequently used PROMs in metastatic CRC trials identified by the search on clinicaltrials.gov for the period between 2010 and 2022.

Table 3 depicts the relationship between patient-reported and clinical outcomes in the 20 published randomized trials included in our review. 

In 5/10 trials (50%), clinical outcomes were reported that favored the experimental treatment. This was supported by patient-reported outcomes favoring the experimental arm [14,23,30,31,32]. In 1/2 (50%) trials, clinical outcomes were reported that favored the standard arm. This was supported by patient-reported outcomes favoring the standard arm [15]. In eight studies, equivalent clinical outcomes were reported between experimental and standard treatment [18,19,21,24,25,26,27,29]. As per primary endpoint, patient-reported outcomes favored the experimental treatment in none (0%), favored the standard treatment in one (14%) [19] and neither favored the experimental nor the standard treatment in seven (86%) [18,21,24,25,26,27,29] trials, respectively.

A discordance between patient-reported and clinical outcome was found in 7/20 (35%) of trials. In five trials, the clinical outcome favored the experimental treatment (in all trials either a new drug was added to chemotherapy or compared to placebo) whereas the patient-reported outcome was equivalent in both treatment arms [13,16,17,20,22]. In one study, the experimental arm (a less intensive maintenance treatment) was not non-inferior to the standard arm (a more intensive maintenance treatment). Therefore, the clinical outcome favored the standard arm. With respect to global QoL, functional scales and several symptoms/items of EORTC QLQ-C30 (fatigue, nausea/vomiting, appetite loss, diarrhea) and EORTC QLQ-CR29 (body image, dry mouth, hair loss, taste, fecal incontinence, sore skin), and EQ-5D, no significant differences were found between the two arms [28]. In two trials, the patient-reported outcomes favored the standard treatment [15,19]. In one study, the experimental arm was not non-inferior to the standard treatment and the time to the worsening of symptoms (FACT-C) and QoL was significantly shorter in the experimental arm [15]. In the other trial, the clinical outcome with respect to the primary endpoint was equivalent to the standard arm, but the experimental treatment worsened the time to QoL deterioration on the physical and cognitive function, global health, fatigue, nausea, appetite and diarrhea scales [19]. 

In all trials, the effect of the experimental intervention on patient-reported outcomes was classified as an effect on both symptoms and other domains (functional scales and global quality of life).

### 3.4. The Role of PROMs in the Determination of the Value of the New Treatment Modalities

A total of 46 trials registered on clinicaltrials.gov used PROMs. Patient-reported outcomes were a secondary or exploratory endpoint in all 46 trials. In three out of 20 publications [18,24,25] (15%) derived from these 46 studies, although mentioned in the results section, patient-reported outcomes were not brought out in the discussion section of the publication nor included in the final assessment and positioning of the new treatment compared to the reference treatment.

## 4. Discussion

In this study, we found that the EORTC QLQ-C30, the EQ-5D and the EORTC QLQ-CR29 were the most frequently used PROMs in randomized trials on metastatic CRC between 2010 and 2022. The use of the EORTC QLQ-C30 and the EORTC QLQ-CR29 increased over time. When comparing patient-reported outcomes to the clinical outcomes in the reported randomized trials, the patient-reported outcomes correlated moderately (60%) with the objective outcome measures of the trials. The quality of assessing PROMs in metastatic CRC trials according to the CONSORT PRO extension has continued to improve in recent years. Compared to a systematic review of randomized controlled trials of CRC treatment describing patient-reported outcomes from January 2004–February 2012, which found that the trend toward the improved reporting of patient-reported outcome information has continued compared to the period prior to 2004, the quality of assessing and reporting patient-reported outcomes has further improved [10]. It is worth noting that the quality of patient-reported outcome reporting in that review was assessed in comparison with the ISOQoL reporting standards which comprised more items [10,12]. In order to be “high quality patient-reported outcome reporting”, trials were required to meet at least 12/18 items (or 20/29 items for trials with patient-reported outcomes as the primary endpoint) and, in addition, three items were mandatory (i.e., baseline compliance, psychometric properties and missing data reported). In the present review, we used the CONSORT PRO extension consisting of five items. Despite these differences, we could retrieve the CONSORT PRO items and compare the results to the findings of the current review. Furthermore, the randomized controlled trials included in the systematic review by Rees et al. did not only comprise systemic therapy for metastatic CRC. However, 67% of the included trials was in advanced/metastatic CRC and 68% of the included trials investigated chemotherapy (and 18% targeted therapy) [9,10]. Therefore, we believe that a comparison between both the current review and the systematic review performed by Rees et al. [10] can provide useful information on the trend in the quality of reporting on patient-reported outcomes in metastatic CRC treatment. 

Although the quality of reporting on patient-reported outcomes gradually improved over the years, still, only 46 trials registered on clinicaltrials.gov between 2010 and 2022 used PROMs, and in a period of more than 10 years, only 20 trials in metastatic CRC using PROMs have been published. It appears obvious to report these data and publications from many of the 120 trials that have not been published yet may still happen. Nevertheless, many of the randomized trials on metastatic CRC registered on clinicaltrials.gov failed to incorporate PROMs in the study protocol or did not report PROMs in the subsequent publications, despite being part of the original protocol highlighting that patient-reported outcomes continue to be underreported in the metastatic CRC literature. Clinicians tend to underestimate patients’ symptoms [33] and patient-reported outcomes are essential to capture information that clinicians are not always able to detect. Although PROMs provide evidence on the effect of interventions on patient symptoms and quality of life, they remain subjective and, hence, prone to bias. PROMs may be affected by internal factors, such as mood, expectations, time and sentiments, and external factors such as treatment context, interactions with the healthcare providers and patients’ socioeconomic situation, which leads to variations in the outcomes. Moreover, patients often tend to overrate benefits and underrate the risks of treatments [34]. In particular, if one treatment arm is associated with an initial deterioration in function, such as systemic therapy in metastatic CRC, followed by a sturdy recovery, this can provide an overall perception of significant improvement. This variability of PROMs can be mitigated by collecting measurements at multiple time points to assess the trajectories of symptom progression and recovery. However, in 10% of the publications included in this review, it was unclear whether this was the case. Furthermore, still only 35% of trials in metastatic CRC state the patient-reported hypothesis and only 60% explicitly state statistical approaches for dealing with missing data. Therefore, the risk of multiple statistical testing and the selective reporting of patient-reported outcomes based on statistically significant results is real. Furthermore, especially in longitudinal analyses, dealing with missing data is challenging and missing data are a potential source of bias. A clear overview of the statistical approaches used facilitates the interpretation of the generated PROM results. Thus, methodological limitations in trials describing PROMs may still hamper patient-centered decision-making about the optimal treatment for metastatic CRC. In the present review, 85% of the trials that used PROMs discussed the patient-reported outcomes in the subsequent publications and included the results of PROMs in the final assessment and the positioning of the new treatment compared to the reference treatment, therefore providing relevant information that will likely support clinical decision-making. Due to the improved quality of collecting and reporting PROMs, the discussion of their clinical significance in publications has improved accordingly. In the aforementioned review [10], only 42% of the trials discussed generalizability issues and 62% of published studies put the findings of the collected patient-reported outcomes in context. Nowadays, this has improved, since 85% of metastatic CRC trials did discuss PROM results in the publications and included them in the final assessment and the positioning of the new treatment compared to the reference treatment. By doing so, these trials discuss the limitations of PROMs used and aid in incorporating PROM data in clinical practice.

Fully understanding the overall effect of a novel therapy on both patient-reported and clinical outcomes will allow the clinician and patient to make decisions that trade off survival, symptoms, functional abilities and quality of life during shared decision-making. Not reporting patient-reported outcomes even when collected does not recognize the efforts of the patients reporting these outcome measures and leads to the loss of valuable information and selection bias regarding the true value of new treatment options in metastatic CRC.

Whilst PROM results provide insight into the impact of an intervention or therapy on the patient, Patient-Reported Experience Measures (PREMs) provide insight into the quality of care during the intervention. PREMs are validated questionnaires that gather patients’ and families’ views of their experience receiving care and are commonly used to measure the quality of care, with the goal of making care more patient- and family-centered [35]. Therefore, in order to determine the true value of a new treatment in metastatic CRC in daily practice and to help guide patient care, PREMs should be, in addition to PROMs and the functional outcomes of clinical trials, a focus for health care providers dealing with patients with metastatic CRC. 

We have found that the EORTC QLQ-C30 is the most widely used quality-of-life questionnaire in metastatic CRC research (followed by the EQ-5D and the EORTC QLQ-CR29). These results are in line with a recent systematic review on randomized controlled trials with PROMs evaluating conventional medical treatments, conducted in patients with breast-, lung-, colorectal-, prostate-, bladder-, and gynecological cancers [36].

Generally, the choice for a certain PROM in a cancer trial reflects conceptual contemplations, such as whether or not to focus on symptom burden only or to focus on broader health domains (or even conduct a multidimensional health-related quality-of-life assessment). The popularity of the EORTC measures and the EQ-5D could be explained by the fact that they are all multidimensional questionnaires that are not limited to physical functioning or somatic symptoms, but also include broad health-related quality-of-life concepts and assess emotional, social and (role) functional aspects of health. Moreover, EORTC QLQ-C30 is easy to use and patients are able to complete this questionnaire in a reasonable amount of time [37]. The EQ-5D questionnaire has gained a lot of interest and is widely used in oncology because its purpose is to support health technology assessment studies [38]. Notably, the EORTC QLQ-CR29, being designed specifically for CRC research, was ranked only third. It is possible that the other questionnaires are more convenient to use and can also provide reliable results. On the other hand, it may also be due to the fact that EORTC QLQ-CR29 was merely later developed than the EQ-5D and EORTC QLQ-C30. The decision to use a certain PROM was usually made several years prior to publication of study results, so this time lag may explain the fact that EORTC QLQ-CR29 was relatively infrequently used. 

To our knowledge, this is the first study on the proportional assessment of the use of PROMs combined with the assessment of the role of PROMs and the assessment of quality of PROM reporting in trials investigating systemic therapy in metastatic CRC. This study has some limitations. Literature searches do have the risk of excluding studies, and studies in languages other than in English were not included, although this limitation likely does not significantly alter the conclusions [39,40]. Furthermore, studies that have been registered in databases of privately and publicly funded studies other than clinicaltrials.gov (such as the International Standard Randomized Controlled Trial Number (ISRCTN) database, EU Clinical Trials Register, Pan African Clinical Trial Registry) may have been missed. Nevertheless, clinicaltrials.gov is by far the largest clinical trial database, and we believe that the conclusions of this review will not be significantly altered by extending the search to other databases. Lastly, we used the CONSORT PRO extension instead of the full checklist. This may lead to a misinterpretation of what constitutes complete reporting and could perpetuate poor reporting practices [41]. However, we transparently showed the number and nature of items of the CONSORT PRO checklist which we included and a brief explanatory text, and how we weighed the specific items to minimize the risk of misinterpretation of what is recommended in a publication of patient-reported outcome trial endpoints.

## 5. Conclusions

The quality of reporting on PROMs in metastatic CRC trials has increased over the past 10 years and the most frequently used PROMs were the EORTC QLQ-C30, the EQ-5D and the EORTC QLQ-CR29. However, in a period of more than 10 years, only 20 trials in metastatic CRC using PROMs have been published. Still, only 35% of trials addressed all CONSORT PRO extension items. Better reporting of patient-reported outcome may translate into a greater impact of study findings on real-world practice. PROM results ought to be an integral part of clinical trial results and the quality of data will be improved when journals demand that they are to be published according to broadly accepted standards and put into perspective alongside the other clinical endpoints of the trials.

In order to guide decision-making regarding the systemic treatment of metastatic CRC based on patient-reported outcomes, using the same PROMs in future trials and reporting and assessing PROMs with high quality will provide for more rigorous evidence generation, and a systematic review on those trials is the next step forward. 

## Figures and Tables

**Figure 1 cancers-15-01135-f001:**
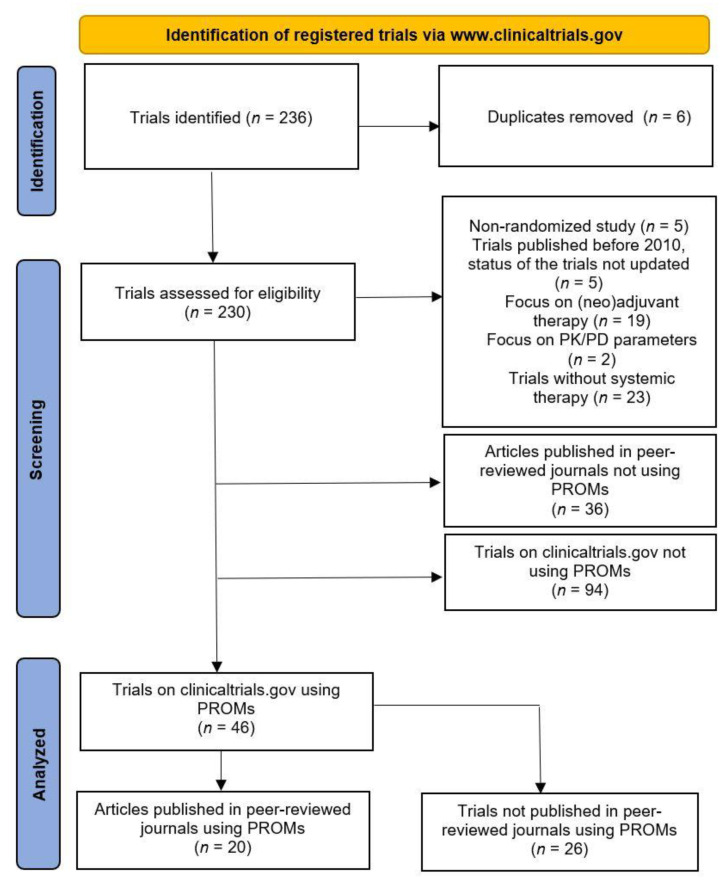
Flow diagram of the literature search and selection process. PK/PD, pharmacokinetics/pharmacodynamics; PROM, patient-reported outcome measure.

**Figure 2 cancers-15-01135-f002:**
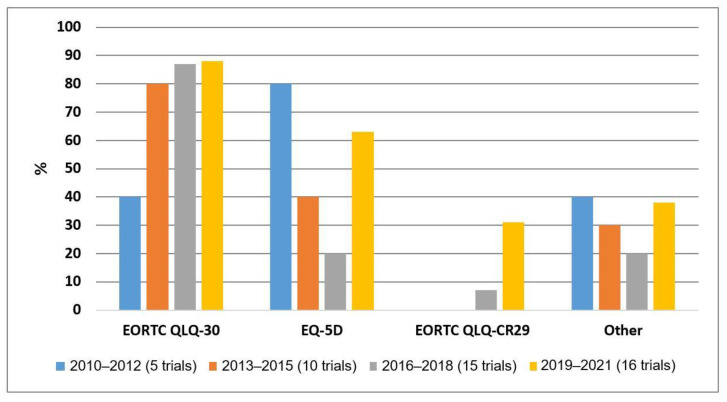
Time trend for the frequency of PROMs used most frequently in metastatic CRC randomized trials between 2010–2022. Percentages are given relative to the number of trials per period that used at least one PROM. Between 2010–2012, five trials were registered; between 2013–2015, 10 trials; between 2016–2018, 15 trials; and between 2019–2021, 16 trials. EORTC QLQ-C30, European Organization for Research and Treatment of Cancer general health status and quality-of-life questionnaire; EQ-5D, EuroQol-five dimension index questionnaire; EORTC QLQ-CR29, European Organization for Research and Treatment of Cancer general health status and quality-of-life questionnaire-Colorectal Cancer Module 29.

**Table 1 cancers-15-01135-t001:** Published randomized trials on systemic therapy in metastatic CRC reporting PROMs.

Publication	*n*	Treatment Line	Primary Endpoint	%PROM Evaluable *	PROMs Used	Separate PRO Publication	Adjusted CONSORT PRO Checklist Score
Bennett [13]	530	2	PFS	89	EQ-5D + VAS	No	5/6 = 0.83
Odom [14]	363	3	PFS	96	EQ-5D, NFCSI -19	No	5/6 = 0.83
Schmoll [15]	1614	1	PFS	84	FACT-C	Yes	3/6 = 0.50
Grothey [16]	753	3	OS	unknown	EORTC QLQ-C30, EQ-5D	No	3/6 = 0.50
Láng [17]	666	1	PFS	94	EORTC QLQ-C30	Yes	6/6 = 1.00
Carrato [18]	768	1	PFS	90	EQ-5D, MDASI-GI	Yes	3/6 = 0.50
Ringash [19]	750	3	OS	96	EORTC QLQ-C30	No	6/6 = 1.00
Siena [20]	505	1	PFS	88	EQ-5D + VAS	Yes	4/6 = 0.67
Hamidou [21]	284	1	DFS	60	EORTC QLQ-C30	Yes	5/6 = 0.83
Quidde [22]	472	1	PFS	88	EORTC QLQ-C30, QLQ-CR29	Yes	6/6 = 1.00
Hickish [23]	333	3	OS	93	EORTC QLQ-C30	No	3/6 = 0.50
Jonker [24]	282	3	OS	unknown	EORTC QLQ-C30	No	2/6 = 0.33
Aparicio [25]	494	1	DCD	68	EORTC QLQ-C30	Yes	2/6 = 0.33
Lacas [26]	410	1	PFS	99	EORTC QLQ-C30	Yes	6/6 = 1.00
Lenz [27]	768	>3	OS	88	EORTC QLQ-C30, EQ-5D	No	6/6 = 1.00
Raimondi [28]	229	1	PFS	92	EORTC QLQ-C30, EORTC QLQ-CR29, EQ-5D	Yes	6/6 = 1.00
Wolstenholme [29]	1103	1	OS	92	EORTC QLQ-C30, EQ-5D, EORTC QLQ-LMC21	Yes	5/6 = 0.83
Hofheinz [30]	760	3	OS	unknown	EQ-5D, EORTC QLQ-C30	No	5/6 = 0.83
Andre [31]	307	1	OS, PFS	96	EORTC QLQ-C30, EORTC QLQ-CR29, EQ-5D	Yes	6/6 = 1.00
Kopetz [32]	665	2, 3	ORR, OS	>83	EORTC QLQ-C30, FACT-C	No	4/6 = 0.67

* Denotes the percentage of patients with evaluable PROM data according to the authors of the published trial. PFS, progression-free survival; OS, overall survival; DFS, disease-free survival; DCD, disease control duration; ORR, overall response rate; PRO, patient-reported outcome; PROM, patient-reported outcome measure; EQ-5D, EuroQol-five dimension index questionnaire; VAS, visual analogue scale; NFCSI-19, National Comprehensive Cancer Network/Functional Assessment of Cancer Therapy Colorectal Cancer Symptom Index-19; FACT-C, Functional Assessment of Cancer Therapy-Colorectal; EORTC QLQ-C30, European Organization for Research and Treatment of Cancer general health status and quality-of-life questionnaire; MDASI-GI, MD Anderson Symptom Inventory for gastrointestinal cancer; EORTC QLQ-CR29, European Organization for Research and Treatment of Cancer general health status and quality-of-life questionnaire-Colorectal Cancer Module 29; EORTC QLQ-LMC21, European Organization for Research and Treatment of Cancer general health status and quality-of-life questionnaire-Liver Metastases Colorectal 21; CONSORT PRO, patient-reported outcome extension to the Consolidated Standards of Reporting Trials (CONSORT) statement.

**Table 2 cancers-15-01135-t002:** Overall level of reporting of patient-reported outcomes according to the CONSORT PRO extension and comparison with a systematic review on trials of CRC treatment describing patient-reported outcomes published from January 2004–February 2012 [10].

CONSORT PRO Extension	Brief Explanatory Text	Present Review (*n* = 20) (%)	Rees et al. (*n* = 66) (%)
**(P1b)** The PRO should be identified in the abstract as a primary or secondary outcome	*Explicitly identifying PROs in the trial abstract will facilitate indexing and identification of studies to inform clinical care and evidence synthesis*	15 (75)	55 (83)
**(P2b)** The PRO hypothesis should be stated and relevant domains identified, if applicable *	*PRO measures may be multi-dimensional and may assess patient status at several time points during a trial. A pre-specified hypothesis reduces the risk of multiple statistical testing and selective reporting of PROs based on statistically significant results*	7 (35)	21 (32)
**(P6a)** Evidence of PRO instrument validity and reliability should be provided or cited if available ^†^	*This information will allow readers to assess the validity, reliability and appropriateness of the PRO being used*	20 (100)	46 (70)
**(P6aa)** Mode of administration, including the person completing the PRO and methods of data collection (paper telephone electronic other).	*Different methods of data collection could lead to potential bias when interpreting outcomes*	18 (90)	20 (30)
**(P12a)** Statistical approaches for dealing with missing data are explicitly stated.	*Missing PRO data is a potential source of bias. A number of methods for dealing with missing data are available with different strengths and limitations which should be described to facilitate interpretation*	12 (60)	37 (56)
**(P20/21)** PRO-specific limitations and implications for generalizability and clinical practice should be discussed	*PRO specific limitations may influence the generalizability of results and use in clinical practice*	17 (85)	32 (48)

* this percentage is calculated by considering all studies (including those explicitly reporting an exploratory evaluation for which this item would rate as Not Applicable); ^†^ In the case of studies using multiple PROMs, we evaluated this as “yes” if at least one PROM was validated. CONSORT PRO extension, patient-reported outcome extension to the Consolidated Standards of Reporting Trials (CONSORT) statement; PRO, patient-reported outcome.

**Table 3 cancers-15-01135-t003:** Relationship between patient-reported outcomes and objective clinical outcome in the 20 selected randomized trials.

		Clinical Outcome		
		Favoring the Experimental Treatment	Equivalent	Favoring the Standard Treatment
**Patient-reported outcome**	**Broadly favoring the experimental treatment**	5	0	0
	**Neutral**	5	7	1
	**Broadly favoring the standard treatment**	0	1	1

## Data Availability

Not applicable.

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
