# Peer review of "Role of Patient-Reported Outcomes in Clinical Trials in Metastatic Colorectal Cancer: A Scoping Review"

_cancers, 2023, doi:10.3390/cancers15041135_

Round 1
Reviewer 1 Report (Previous Reviewer 1)
Thank you for giving me the opportunity to review again this manuscript. I appreciate the work made from the authors in response to the reviewer’s comments. This revised version is clearer, and the added details have increased the quality of the paper. Although these improvements, I think that further issues should be addressed by the authors before publication.
Major
Line 34: “A total of 56/176 registered trials were published”. I can’t find this information on the manuscript.
As the use of CONSORT-PRO represents one of the most important parts of this work, some details should be added. For example, it is important to add that the CONSORT-PRO consists of 14 items, 5 PRO-specific extensions (used in this manuscript) and 9 PRO-specific elaborations. Furthermore, the authors of the checklist recommended to use the full checklist (see Mercieca-Bebber Qual Life Res. 2022 Oct;31(10):2939-2957) as studies using incomplete or modified checklists may confuse readers about what is recommended in a publication of PRO trial endpoints. This last aspect represents a limitation of this study that authors should add in the discussion.
Lines 141-146: the wording that describe the criteria used to evaluate the PRO direction corresponds exactly to the wording used in the legend of Table 1 of the paper of Rees et al. I would suggest the authors to rephrase the sentence and to add that the criteria used in their work are based on the ones used by Rees et al.
In the Table 1, before the last column I would add a column reporting the information on whether the study has a secondary publication focused on PRO or not. This would help interpreting the score as, if the results are in line with other studies, it is expected that the trials with a minor score are those without a PRO publication. I would also suggest the authors to add in the results the mean score of the studies with and without a secondary PRO publication.
When the authors compared the frequency of studies with higher quality of reporting found in their review with the frequency reported in the review of Rees, they are making a comparison stemming from different criteria. The “high-quality” criteria of the review of Rees were based on the ISOQOL checklist and required that at least 12/18 items (20/29 for studies with PRO as primary endpoint) were satisfied and, in addition, three items were mandatory (i.e. baseline compliance, psychometric properties, and missing data reported). The criteria used by the authors, instead, is the satisfaction of the 5 items of the CONSORT-PRO extension. It is important to add this information for the readers.
Table 2: item P6aa. I think that the correct N and percentage for Rees et al is 20 (30). Please check.
In the conclusions, the authors stated that 30% of trials reported PROs with high-quality. As in the methods are not described the criteria to define a study as having a “high-quality” reporting, and as the authors did not use the full checklist, I would suggest to just say that 30% of trials addressed all the CONSORT-PRO extension items.
Minor
Line 18: I would add “their disease” after “associated with”, as PROMs capture HRQoL associated both to the disease and treatments.
Line 75: “, including Quality of Life (QoL),”. I would move this part in line 77, after “other aspects”.
I would suggest the authors to further check the inconsistencies in the use of the terms PRO or PROM. For example, in line 87 instead of “PROM questionnaires” I would use either “PRO questionnaires” or just “PROMs”, as PROM are questionnaires.
Line 255: add the % after 37.
Line 336: 60% should be 65%?
Line 447: 30% should be 35%?
Author Response
Please see the attachment.

Reviewer 2 Report (Previous Reviewer 3)
The manuscript has been improved by the adaptations, no further comments.
Author Response
Please see the attachment.

Reviewer 3 Report (New Reviewer)
The outcome from this study presenting EORTC QLQ-C30, the EQ-5D and the EORTC QLQ-331 CR29 as the frequently used outcome measure for metastatic CRC is interesting. The work presented by Maring et al is excellent.
However for metastatic colorectal cancer treatment the author could discuss and include some of the CRC biomarkers such as carcinoembryonic antigen (ACS Appl. Mater. Interfaces 2022, 14, 9, 11078–11091) in their discussion for future scope of therapy.
Author Response
Please see the attachment.

This manuscript is a resubmission of an earlier submission. The following is a list of the peer review reports and author responses from that submission.
Round 1
Reviewer 1 Report
Comments on the manuscript: Role of patient reported outcomes in clinical trials in metastatic colorectal cancer: clinical decision aid or just nice to have?
Thank you for giving me the opportunity to comment on this manuscript. The authors performed a scoping review on the use of PRO in phase III clinical trials started between 2010 and 2021 evaluating systemic therapy in patients with mCRC. The objective of the study is interesting and important. However, there are some major issues that should be addressed and that may limit the interpretation of the study results.
MAJOR
In my opinion, one of the main limitation of this review regards the evaluation of the quality of PRO reporting. According to what reported by authors, this represents one of the main outcome of this scoping review, as the increasing of the quality of PRO reporting was highlighted in the abstract, discussion and conclusion. However, no sufficient information supporting this conclusion was provided by the authors. In the methods there is no information about how authors assessed the quality of PRO reporting. Did they use one of the internationally endorsed checklist for assessing quality of PRO reporting (e.g. CONSORT-PRO, ISOQOL)? If yes, did they use all of the items or a selection? How did they calculate the quality of PRO reporting for each study? If no published checklist was used, was an ad-hoc checklist used and why? Also, a table reporting all the percentages of studies endorsing each item of the checklist used should be reported, and results should be described in the Results section.
General comment: I suggest the authors to pay attention not to confuse the terms PROM and PRO. The first one refers to the instruments (generally questionnaires) for measuring PRO. The authors used throughout the manuscript only the term PROM, also when it would be better to use PRO. For example, in the last sentence of the abstract “PROM” should be replaced with “PRO”. Authors should check the manuscript and correct these inconsistencies.
Methods are a bit confusing and should be better organized to facilitate the comprehension of the steps performed by authors. It should be better described separately the two steps of the search, by reading the chapter it seems that after searching in clinicaltrials.gov a screening of abstracts and full-text was performed there, but I suppose this was made on Pubmed. It is not clear if the authors searched for studies or for publications, and this is confusing also in the flow diagram, where both terms are used (e.g. in the excluded record, did the authors excluded the studies or the publications?). Did the authors searched first the studies on clinicaltrials.gov, then searched for the publications? Why not searching directly on Pubmed? Sometimes in clinicaltrials.gov the status of the trials are not updated and publications are not automatically indexed. I suggest the authors to describe the process in the chronological order by which it was performed.
Authors should better describe the criteria used to evaluate the comparison between clinical outcomes and PROs. For clinical outcomes: which endpoint where considered to evaluate the direction (i.e. better, equal, worse)? Only the primary clinical endpoint? Only survival? All the endpoints? Which parameter was used to determine the direction? The p-value, the HR, etc? For PRO: did the authors considered all the outcomes from all the instruments? All the scales of the instruments? They considered the statistically significant scales, the clinically meaningful scales or both?
At the beginning of the Results section I would suggest the authors to better summarize the results of the searches, to better guide the reader afterwards. For example, the search in clinicaltrials.gov identified xx trials, of these xx measured PRO, of these with PRO xx are still ongoing and xx are concluded, of the concluded studies with PRO xx published PRO results in paper.
In the simple summary authors reported to have searched for studies started from 2010, but they eventually selected 5 trials published before 2010. Please clarify this.
Table 1: to what the authors refer with the column “% PROMS completed”? Only the PROMS at baseline, at follow-up, all the time-points? What does it mean “PROM assessment clear and repeated” and which were the criteria used to evaluate these aspects? Which where the criteria used to evaluate if PROM were discussed? Just mentioned in the discussion, re-state the results, or discussed critically? Please add a legend with all the specification (also in the methods should be added this information) and the definition of the acronym. Also, in the description of the instruments used, for Bennett it was reported EQ-5D and VAS as they were different instruments. In that study only the EQ-5D was used and the VAS scale is the one of the EQ-5D instrument.
Figure 2 did not depict data in the best way in my opinion, as results are very difficult to read and interpret. I would suggest to the authors to report the data the same way as reported in Figure 1 of the reference #36 (Giesinger et al.), as it may better shows the trends of the use of PRO instruments over-time. Also, please specify if numbers on y-axis are absolute number or %. Finally, the title of the Figure reports “trials 2010-2012” but in the figure are also reported data before 2010. Please correct this inconsistency.
I would not start the discussion highlighting that “only 31 trials were published” because it is not the main results and it is reported as a negative outcomes. It would be negative if all the trials would be concluded but as many of them are still opened it is normal that no publication has been published yet. Without this specification this information is biased.
Line 206: “most PROMs correlated well”, actually is 52% so half of them. I would tone-down this statement.
Line 228: actually in the original review cited (ref #33), the 29% of trials discussed the clinical significance of PRO results. The % of trials discussing PRO results in the context of other trials outcomes were 64%.
Lines 239-240: I would suggest the authors to also add the fact that clinicians tend to underestimate patients’ symptoms, as well described in literature (e.g. Di Maio et al. J Clin Oncol. 2015 Mar 10;33(8):910-5). Without this addition, the information conveyed by this sentence is that patients are more prone to incorrectly report their symptoms while evidence demonstrated that PROs are essential to capture information that clinicians are not always able to detect.
MINOR
Abstract, line 33. “PROMs are collected in all publications”. It should be better to say “PROs are reported in all publications”, as PROs are “collected” in the studies, not in publications.
Was this study conducted according to the extension for scoping reviews (PRISMA-ScR)? If yes, please reported it.
Authors should explain the rationale for having decided to start the search from 2010.
Line 161: “Only 21 of these studies resulted in publications of the PROM results”. The number of references after this sentence are 24, not 21. Please clarify why.
Line 196: What did the authors mean with “tertiary” endpoint? Exploratory?
Line 212-214: it is not completely clear this sentence. Please rewrite it.
Line 225: I think that the correct reference should be the #33 instead of #28?
Author Response
See attached pdf file

Reviewer 2 Report
The manuscript entitled “Role of patient reported outcomes in clinical trials in metastatic 2 colorectal cancer: clinical decision aid or just nice to have” by Maring et al have shown the systematic review on Patient Reported Outcome Measures (PROMs) on phase III clinical trials between 2010 and 2021 those were evaluated 21 systemic therapies in patients with metastatic colorectal cancer(mCRC). Authors addressed the pertinent questions for treatment verses publication and comparative analyses of PROMS. Authors revealed that 31% of the enrolled randomized studies on systemic 202 therapy in mCRC in clinicaltrials.gov between 2010 and 2022 was published and that only 203 35% of these publications reported PROMs. Although and 11% of published trials failed to explain the relevance of PROMs to the overall trial findings and often did not incorporate PROMs into subsequent treatment recommendations. This is very interesting report to understand the clear picture of the translational research and publication to improve the research & Practices. Manuscript is written well. Authors may approach more appropriate way to simplify the explain the data and presentation for readers such as Table 1 shown treatment line 1,2,3. Authors may explain in figure legends in details.
Author Response
See attached pdf file

Reviewer 3 Report
The study focusses on an interesting and clinically relevant topic.
Main points:
1. The authors conclude that the quality of the PROMs is assessed and is improved over time.
However, the terminology used in evaluating the quality of assessment can be somewhat confusing. I would say that you primarily focus on the quality of describing the administration and data collection. This is an important quality evaluation, but a correct application and evaluation of PROMs (especially in an mCRC setting with HRQoL assessment) also very much includes how loss to follow-up, decline of PROM completion with multiple measurements etc. is handled. This is very relevant to the second and third question that you state at the end of the introduction (the relationship between patient reported and clinical outcomes & the role of PROMs in the assessment of the value of new treatment modalities and strategies)
To make the statements regarding the quality improvement of the PROMs a direct assessment of quality and comparison with the results from 10 year ago in a similar trial set should be made.
2. Considering this review focuses on mCRC patients (and includes quite a few second-/third-line studies), I think more emphasis should be placed on the importance of repeated measures (and the quality thereof), especially concerning HRQoL questionnaires. This also means that the quality of assessment (including table 3) should be examined more thoroughly than is the case in the current manuscript. E.g. a median 90% completion rate of at least 1 PROM is not very relevant in a HRQoL situation examining a line of treatment.
3. The authors used clinical trials.gov to select RCTs, why is this method choosen? Why not a pubmed search? Are the results representative for non-registered trials. The results are compared to a systemetic review with a different scope, this comparison is difficult given the difference in inclusion criteria.
Comments per section:
Simple summary:
- Line 20: Scoping review is not layman language
- Line 22: "quality has improved over the last decade" How?
- Line 25: "making it still difficult" --> consider rephrasing
- Line 26 "patient values" consider rephrasing to layman terms (e.g. "relevant for individual patients")
Abstract:
- Line 30: "qualitative analysis" --> the word qualitative isn't mentioned after the abstract and I don't think you have performed a qualitative analysis. I think you performed a two-stage quantitative analysis, where the second part is of a more qualitative nature, but still assessed quantitatively
- Line 34: "The EORTC (..) was most frequently used". Consider adding a proportion or number to this.
- Line 36 "In 16% of publications" is somewhat confusing. Consider something like: "in 3/19 (16%) of publications that use PROMs".
- Line 37 "PROMs are underreported in studies in mCRC". This statement requires importance of reporting PROMs, which you formally didn't analyze in this review. Consider rephrasing to something like "xx % / a minority of studies"
- Line 38 "The qualitative of reporting PROMs is generally high" --> this is not backed by the results section of the abstract and only briefly backed in the results section. Consider adding something to the abstract and results section or rephrase this statement
Introduction:
- Line 47 "The incidence of CRC has been ..." --> needs a source.
- Line 65 "These datapoints are different from the clinical outcomes" --> Considering to provide examples/add more backing to this statement, also because the abstract now concludes that PROMs are underreported
- Line 72 "in clinical trials evaluating ... in patients with metastatic CRC" --> consider to provide some more explanation why you focus on mCRC and/or clinical trials.
Methods section:
- Line 79 and further: consider rephrasing this to something like "keywords X, Y and Z and their synonyms" and refer to an appendix, especially since this is a scoping review
- Line 116: "subsequently, we correlated the patient reported outcomes to the clinical outcomes" --> consider to provide a bit more explanation (and expand table 3 as well, see results section). Line 118 and further I think refers to how you correlate outcomes, so this could be rephrased to one paragraph
Results section:
- The buildup of the results section could be improved by first explaining the different questionnaires first (line 194 and further + add the text about all the different questionnaires to this section), followed by the section about PROMs use over time (line 155 and further)
- As a general statement, numbers can be somewhat confusing. E.g. Line 160 The 45 studies that used PROMs cannot (easily?) be retrieved from the flowchart. Adding an extra block/section to the flowchart could improve interpretation
- Line 146 "all publications involved patients treated in thrid line and beyond" --> this was confusing to me, since prior to this statement, several first-line and second-line studies were described
- Line 148 "a median of 90%" (range 60-99%) --> I think the range refers to the range of medians? Consider adding this
- Line 152 "Table 1 provides an overview of the publications" --> consider to start the paragraph describing the 19 studies (start at line 135 (?)) with this.
- Figure 1: flowchart numbers do not add up. 230 - 5 - 19 - 2 - 23 - 36 - 121 = 24, not 19. Also, consider to add a block to illustrate the 45 studies that you evalute in line 160 and further
- Table 1: % PROMS completed needs a bit more explanation. I think this refers to "at least one PROM completed"
- Figure 2: stacked bar might not be the best choice for this particular representation, because some studies use multiple questionnaires. Also, the total number of studies assessed could be added to this figure to improve interpretation
- Table 2: consider adding a bit more explanation to this table. E.g. I would be interested in which questionnaires were used for which endpoints, even though numbers will become relatively small. If not, you could consider to describe the cell "Unfavorable/Equivalent (n=2)" in a bit more detail, because I think in some way this backs the statement in the abstract "PROMs are underreported in studies in mCRC" a bit, even though I'd say that then still more information is needed (e.g. were multiple questionnaires used, reasons for unfavorable questionnaire responses)
Discussion
- Line 202/204/210/211: consider adding the absolute number of studies to the shown percentages again.
- Line 208 "The quality of assessing PROMs in mCRC trials..": I think this statement needs more backing, because quality of PROM assessment is not only dependent on administration and methods of collecting data, but e.g. patients not completing questionnaires should also be assessed, especially in cases where follow-up questionnaires are warranted (and loss to follow-up/longitudinal analysis is known to be challenging)
- Line 219 and further "in contrast, a systematic review from January 2004 ..." --> consider adding more explicitly whether this review only described studies that did include PROMs. This helps to put the proportions into context.
- Line 226 "due to the seemlingly improved quality of collecting...": What is the motivation for the seemingly improved quality? Systematic review vs your scoping review with more recent studies?
- Line 231 "however, there is still room for improvement" --> consider directly and explicitly stating what can be improved
- Line 232 "and, still, many of the randomized trials on mCRC..." consider making this is a separate statement
- Line 249 "The assessment of treatment efficacy requires ..." --> (part of) this paragraph could be added to the introduction to underline the importance of the use of PROMs.
- Line 290 "To our knowledge" --> Consider adding more strenghts (e.g. proportional assessment of PROM use AND the role of PROM in the trials AND the quality of PROM use)
- Line 291 "This study has some limitations" --> Consider adding something about why you conducted a scoping review instead of a systematic review (or add that to the introduction)
Conclusion:
- Line 301 "The quality of reporting PROMs in mCRC trials --> Considering this is the first statement in the conclusion, I think the direct assessment of quality (and comparison with the systemic review assessing RCTs from 2004 to 2022) should be expanded somewhat. On a sidenote I think the current conclusion would be that the quality has increased compared to studies published until 10 years ago, since you evaluated all included trials as high.
- Line 306 "but the importance attributed to PROMS in determining ..." --> this is an important statement, but I think the current results section could emphasize this more than it is now. E.g. by expanding table 2 with more (textual) detail.
- I agree with your final statement. Consider also adding an advice for future (systematic) reviews. Citation from the 2018 paper by Munn et al focusing on the difference between scoping and systematic reviews: "Scoping reviews are useful for examining emerging evidence when it is still unclear what other, more specific questions can be posed and valuably addressed by a more precise systematic review" https://bmcmedresmethodol.biomedcentral.com/articles/10.1186/s12874-018-0611-x
Writing comments
- Writing of numbers is not consistent. E.g. Line 130 and further.
- Figure and table headers are not entirely consistent (some above, some below)
- Line 34 typo "seriallly"
- Line 66 consider "is becoming" for both statements
- Line 155 PROMs Abbreviations are added to figure 2 (which is after table 1 in the text), not to table 1. Abbreviations in table 1 are also not yet explained in the text. Advise to add abbreviations to both figure 2 and table 1, or replace them below table 1.
- Line 228 "33" probably refers to source?
Author Response
See attached pdf file
